# Bring dying at home: What facilitates and hinders home-based end-of-life care for people living with dementia?—A systematic review and meta-ethnography protocol

**Guo Yin\*, Divya Sivaramakrishnan, Leah Macaden**

Nursing Studies, School of Health in Social Science, University of Edinburgh, Edinburgh, United Kingdom

\* G.Yin-1@sms.ed.ac.uk

## Abstract

### Background

Although home-based end-of-life care is more in line with the preferences of people living with dementia, operationalizing this ideal remains challenging. Many people living with dementia are still unable to die at home or receive end-of-life care at home. This review aims to apply meta-ethnography to synthesize existing qualitative studies, to identify the facilitators and barriers of home-based end-of-life care for people living with dementia.

### Methods

This review will use the meta-ethnography method to systematically synthesize and analyze qualitative studies. The seven stages described by Noblit and Hare (1988) will serve as the framework for this review. The systematic literature search will comprehensively cover the following databases: PubMed, MEDLINE, EMBASE, Cochrane Library, PsycINFO, CINAHL, and Web of Science. Inclusion criteria are: (A) qualitative research design; (B) participants are people living with dementia, family caregivers, or healthcare providers; (C) discussion of barriers and facilitators affecting home end-of-life care for people living with dementia; (D) original peer-reviewed studies in English. The included studies will be quality assessed using the CASP quality assessment form. The entire research process will refer to the meta-ethnography reporting guidelines (eMERGe) and the PRISMA statement to ensure the scientific and systematic nature of the results.

### Discussion

This review will synthesize and analyze the results of different qualitative studies, transforming different perspectives through an iterative process of comparison, translation, and synthesis to generate new insights, and will form a comprehensive and insightful interpretive framework. This will promote a more comprehensive and in-depth understanding of the facilitators and barriers to the implementation of home-based end-of-life care for people living with dementia. In addition, the results of this review will guide the development and

**Data Availability Statement:** No datasets were generated or analysed during the current study. All

relevant data from this study will be made available upon study completion.

**Funding:** The author(s) received no specific funding for this work.

**Competing interests:** The authors have declared that no competing interests exist.

improvement of home-based end-of-life care interventions for people living with dementia, and guide policymakers and practitioners to optimize relevant policies and services.

## 1. Introduction

Dementia is an unpredictable, progressive, and incurable neurodegenerative disease that gradually impairs cognitive functioning and the ability to perform daily activities [1], ultimately leading to death [2]. Dementia has become a common cause of death in the elderly population [3]. With the aging population, the prevalence of dementia is projected to reach 152 million by 2050 [4]. This large dementia population means that society will bear a huge number of deaths [5]. In this context, how to effectively provide high-quality end-of-life care services has become crucial. End-of-life care is a comprehensive medical and supportive service care provided by a multidisciplinary team (including doctors, nurses, social workers, chaplains, or spiritual support personnel) for patients with life limiting or terminal conditions who have a life expectancy of less than 12 months and their families; its goal is not only to alleviate the physical suffering of patients, but also includes comprehensive care at psychological, spiritual, and social levels to improve the patient's quality of life in the last days [6, 7].

End-of-life care was originally designed around cancer patients, but people living with dementia are now one of the fastest-growing groups of hospice registrants, accounting for nearly half of all hospice registrations [8, 9]. However, the increasing number of individuals receiving end-of-life care is placing growing pressure on healthcare systems in terms of resources and costs [10, 11]. End-of-life care services is essentially high-cost and labor-intensive service, particularly when provided in an institutionalized setting such as a hospital, nursing home, or hospice [12], that is usually accompanied by high operating costs, including facility maintenance, professional salaries, and medical equipment costs. Consequently, balancing the enhancement of end-of-life care service quality with cost control has emerged as a critical issue.

Given this context, the home is increasingly seen as a potential setting in the end-of-life care of people living with dementia [13–15]. Home-based end-of-life care utilizes the support services of the health system to enable terminally ill patients to die with dignity in their familiar home environment [16]. Compared to institutionalized end-of-life care, home-based end-of-life care is more cost-effective [12, 17]. Although home-based end-of-life care also requires costs, it can reduce expenses through the effective integration of family and community resources and avoiding the high cost of long-term care facilities or hospital stays [18, 19]. This optimal allocation of resources not only reduces the care costs and economic pressures on patients and families, but also makes more efficient use of limited medical resources and reduces the burden on the public health system.

In addition, home-based end-of-life care not only addresses the high costs associated with institutional care, but also aligns with the growing recognition of the importance of providing personalized and compassionate care in a familiar environment. "Home" represents more than just a living space; it also carries deeper meanings such as safety, familiarity, autonomy, and intimate support [20]. Therefore, dying at home can preserve the patient's sense of normalcy, choice, dignity and comfort [21]. Dying naturally and peacefully surrounded by family at home has been widely recognized as the ideal death [22, 23]. An increasing number of people living with dementia have also expressed a desire to spend their final days at home [24]. Studies have also established that remaining at home during end of life is preferable for most

people living with dementia and their families [25, 26]. This is because the familiarity, stability and predictability of the environment are considered important for people living with dementia. As cognitive functions decline, sensitivity to external changes and unfamiliar interactions increases, and allowing patients to die in a familiar home can protect them from the distress and confusion associated with environmental changes, and avoid the feelings of powerlessness and displacement that might arise when they are transported to a care facility [25]. Furthermore, compared with institutionalized end-of-life care, home-based end-of-life care generally provides people living with dementia and their families with greater comfort and satisfaction [27].

Unfortunately, while a home-based end-of-life care for people living with dementia would be more in line with their preferences, the reality is that people living with dementia often do not receive end-of-life at home or do not die at home, and the majority of people living with dementia are more likely to die in hospitals or long-term care facilities, with the trend of institutionalized deaths continuing [28–30]. Furthermore, people living with dementia are more likely to die in long-term care facilities than those with any other diagnosis [31]. End-of-life care at home remains out of reach for many people living with dementia [24, 31].

Multiple barriers and challenges prevent terminally ill people from receiving end-of-life care at home, as demonstrated by previous studies [22, 32, 33]. Home-based end-of-life care makes the home an extension of the hospital, transferring care and responsibility to the patient's family [34]. People's desire to avoid burdening their families is the main reason they choose to receive end-of-life care outside of the home [22]. Although families wish to care for their dying loved ones at home [32], maintaining continuity of care can be challenging [35]. Inadequate preparation of family caregivers, difficulties in accessing professional support, and lack of adequate support often result in terminally ill people having to be admitted to the hospital and failing to pass away peacefully in the place they desire [32, 33]. Furthermore, the involvement of healthcare professionals can undermine the familiarity and comfort of "home" and make it "institutionalized" [32]. Thus, the provision of end-of-life care in the home raises an important paradox: is home still home? [22] Terminally ill people and their families may have psychological resistance when the nature of the home changes [34], leading them to prefer to receive care elsewhere [35].

Compared with other terminally ill people, people living with dementia face more complex challenges and barriers when receiving end-of-life care at home due to their complex physical and mental symptoms [36]. Unfortunately, existing reviews have not fully captured the complex experiences and specific challenges faced by this population in the home-based end-of-life care process. While there is a growing number of studies in this field [37–39], only one review has specifically explored the barriers and facilitators of dying at home for people living with dementia [36]. Existing review evidence has mostly focused on facilitators and barriers to end-of-life care for people living with dementia, often neglecting the critical context of the home environment [40–42]. Consequently, the absence of a systematic and comprehensive review and analysis of the actual facilitators and barriers related to home-based end-of-life care for people living with dementia results in fragmented findings and scattered evidence. This fragmentation contributes to a significant lack of clarity in the field, making it difficult to effectively address existing blind spots and shortcomings when designing and providing home-based end-of-life care services for people living with dementia. To effectively implement home-based end-of-life care services for people living with dementia, it is crucial to clearly identify the various facilitators and barriers that affect implementation. Moreover, as home-based end-of-life care models continue to evolve globally, new research findings may reveal additional effective strategies and implementation challenges for facilitating end-of-life care for people living dementia [38, 43, 44]. It is therefore crucial to update systematic reviews with

the latest studies to ensure that the evidence base remains current and accurately reflects the dynamic changes in the actual practice of home-based end-of-life care for people living with dementia.

This review will focus on synthesizing qualitative research evidence, as qualitative research has a unique advantage in identifying these facilitators and barriers during the implementation process, providing a deeper understanding of the complex emotions, decisions and challenges stakeholders experience in the home-based end-of-life care process for people living with dementia [45]. **Meta-ethnography** is a comprehensive and rigorous qualitative synthesis approach [46]. Its core is to generate new insights and theories by "translating" and integrating findings from multiple qualitative studies, rather than simply summarizing research results [46, 47]. This method is particularly suitable for dealing with complex, cross-cultural or cross-contextual research problems. Because it can integrate understandings of the same problem from different research backgrounds and perspectives, make up for the cultural and contextual factors that are difficult to capture in a single study, and weaken the influence of individual studies' subjectivity, revealing a deeper understanding and commonality [48]. Compared with traditional narrative reviews, meta-ethnographic reviews provide more methodological descriptions and higher-order interpretations [47], which can provide scientific basis and theoretical support for evidence-based policies and practices in the field of health and social care, and guide decision-making and policy making [49].

Therefore, when exploring the facilitators and barriers to home-based end-of-life care for people living with dementia, meta-ethnography can provide an effective way to develop a comprehensive and insightful interpretive framework by systematically integrating and analyzing the results of different qualitative studies, helping to reveal the facilitators and challenges to the implementation of home-based end-of-life care services for people living with dementia in different environments and cultures.

### Aim

This review aims to conduct a meta-ethnographic synthesis of qualitative research findings to systematically integrate the perspectives, attitudes, and experiences of people living with dementia, family caregivers, and healthcare providers to deeply explore the key facilitators and barriers affecting home-based end-of-life care for people living with dementia. This review will also construct an interpretive framework that systematically presents the main facilitators and barriers of home-based end-of-life care for people living with dementia and present the interrelationships and interactions between these factors in a structured manner.

### 2. Methods

This protocol has been registered on the International Prospective Register of Systematic Reviews (PROSPERO) (CRD42024578005). This review will use a meta-ethnographic synthesis approach to systematically review qualitative studies [46], and the seven phases of meta-ethnography described by Noblit and Hare (1988) [46] will serve as a framework for this review: (1) getting started; (2) deciding what is relevant to the initial interest; (3) reading the studies; (4) determining how the studies are related; (5) translating studies into one another; (6) synthesizing translations; and (7) expressing the synthesis (See Fig 1). This process is iterative, and the various stages are not discrete but may overlap and run in parallel [46, 50].

This review report will also be presented in a structured manner using the Meta-Ethnography Reporting Guidelines (eMERGe) [51] and the PRISMA Statement Guidelines [52] to ensure the clarity and completeness of this review report. Specifically, the eMERGe guidelines will guide this review to provide a detailed description of the seven stages of the meta-

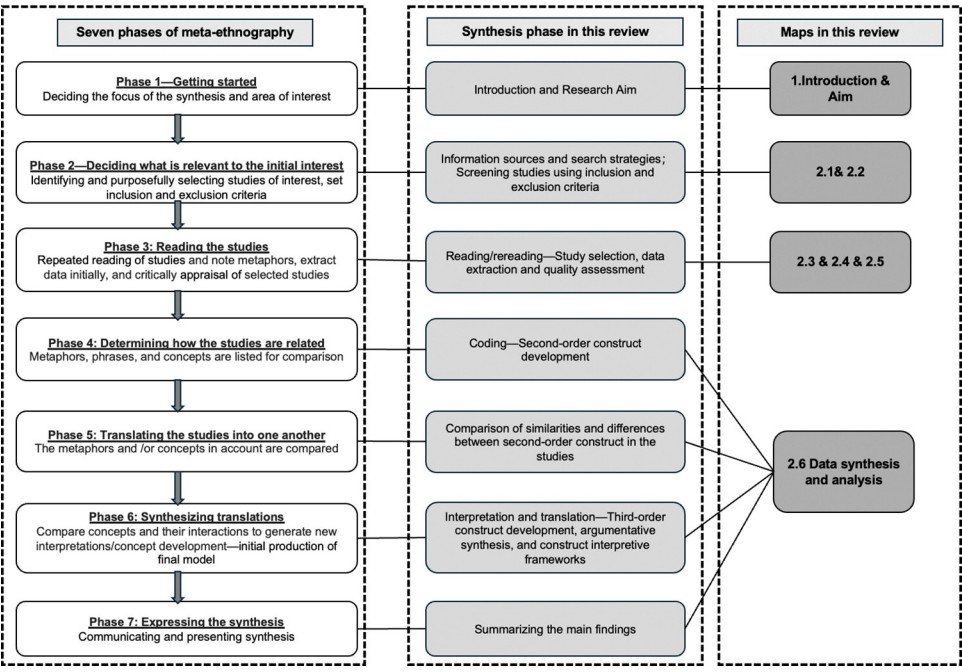

**Fig 1. Noblit and Hare's (1988) seven phases of meta-ethnography.**

ethnographic process, including study selection, data extraction, translation, and synthesis. Meanwhile, the PRISMA statement guidelines will be utilized to ensure that the methodological rigor, transparency, and reliability of this systematic review, including clearly defined research questions, comprehensive literature searches, selection criteria, data extraction methods, and result reporting. The integration of these two guidelines will help present the process and results of this review in a more systematic way.

## Phase 2: Deciding what is relevant to the initial interest

**2.1 Information sources and search strategies.** The systematic literature search for this review will comprehensively cover the following databases: PubMed, MEDLINE, EMBASE, Cochrane Library, PsycINFO, CINAHL, and Web of Science. The search strategy will employ a comprehensive combination of MeSH terms and various free-text terms tailored to each database. The search terms will include concepts related to dementia, end-of-life care, home settings, barriers, facilitators, and qualitative research. To ensure the inclusion of all relevant studies, a variety of free-text terms will be utilized, tailored to the characteristics of each database. The free-text search include different spellings, synonyms and related phrases to ensure that no important studies are missed. Boolean operator (such as AND, OR, NOT) will be constructed based on relevant concepts to ensure a systematic search strategy. Table 1 presents a detailed summary of the search terms. The MEDILINE search strategy is also provided in S1 File. This search strategy will be adapted based on the syntax and subject headings of other databases.

In addition, the reference lists of all included studies will be systematically reviewed to identify relevant studies that may not have been captured initially. Furthermore, citations will be manually searched using Google Scholar to identify any potentially overlooked studies. The search will not be restricted by publication date.

**Table 1. Search terms.**

| | |
|---|---|
| **Participant** | (Dementia OR Alzheimer* OR "Cognitive impairment") And (People OR Patient* OR "Older adult*" OR Individual* OR Caregiver* OR "Family caregiver*" OR "Care provider*" OR "Health personnel*" OR "Health worker*" OR "Health professional*" OR Nurs* OR Physician* OR "Hospice worker*" OR "Social worker*" OR "Stakeholder*") |
| **Intervention** | "End-of-life care" OR "Terminal care" OR "Palliative care" OR Hospice OR Dying OR Death OR Die* |
| **Settings** | Home OR Homes OR Home-based OR Home care OR Home setting* OR House* OR Residence OR Domic* OR Dwelling* |
| **Outcome** | Challeng* OR Barrier* OR Hinder* OR Obstacle* OR Problem* OR Block* OR Deter* OR Difficult* OR Stop* OR Facilitat* OR Enabl* OR Support* OR Allow* OR "Conducive" OR Prevent* OR Experience* OR Perspectiv* OR View* OR Perception* OR "Coping strateg*" |
| **Study design** | "Qualitative research" OR "Qualitative study" OR Phenomenolog* OR "Case stud*" OR "Narrative analys*" OR "Focus group*" OR "Semi structured interview*" OR "Grounded theory" OR "Mixed stud*" |

**2.2 Inclusion and exclusion criteria.**   *2.2.1 Inclusion criteria.* **Participants.** People with any type and stage of dementia; family caregivers or healthcare providers (e.g. doctors, nurses, psychologist etc.).

**Phenomena of interest.** Describes the facilitators and barriers to end-of-life care at home for people living with dementia from a variety of perspectives, including the experiences and views of stakeholders (people living with dementia, health care workers, and family caregivers).

**Outcomes.** Facilitators and barriers to home-based end-of-life care for people living with dementia. Facilitators refer to conditions and circumstances that are actually or perceived to be conducive to the implementation of end-of-life care at home; Barriers refer to conditions and circumstances that actually or perceived to have a negative impact on the provision of end-of-life care at home [36], including psychological and emotional influences of family caregivers, ethical considerations, resource factors, policy influences, and sociocultural factors.

**Study design.** Qualitative research, including but not limited to ethnography, phenomenological research, focus group research, etc. Mixed methods research with an extractable qualitative component, but only the qualitative component of mixed methods research will be considered.

**Publication type.** to ensure the accuracy and applicability of the results, only full-text studies published in peer-reviewed journals will be included.

*2.2.2 Exclusion criteria.* **Non-English literature.** The main reason for excluding non-English literature is that language barriers may lead to misunderstanding and inaccurate interpretation of research findings. Resource limitations also make it impossible to use formal translation services.

**Study design and publication type.** Mixed methods studies that fail to adequately separate qualitative data, quantitative studies, pilot studies, case reports, conference abstracts, conference papers, review articles or literature reviews, and grey literature will be excluded.

**Phenomena of interest.** Studies that focus only on general end-of-life care issues and do not specifically explore the context of home end-of-life care, studies that do not clearly identify people living with dementia as the research subjects, or studies that do not explore or analyze specific facilitators, barriers, or challenges of home-based end-of-life care for people living with dementia will be excluded.

## Phase 3: Reading the studies

**2.3 Study selection.**   The search results will be exported to Covidence software [53], a system that will automatically identify and remove duplicate records to ensure the effectiveness

and accuracy of subsequent review processes. Two reviewers will independently screen the titles and abstracts of the studies according to the established inclusion and exclusion criteria. For studies that meet the eligibility criteria in the preliminary review, two reviewers will conduct a full-text review independently to further confirm whether they are suitable for inclusion. Any disagreements between the two reviewers will be addressed through discussion; if consensus cannot be reached, a third reviewer will be invited to make the final decision. The reasons for exclusion of studies from the full-text screening will be provided and documented in detail by Covidence and will be reported in a table format in the review report [53]. The study selection process and its outcomes will also be visually represented using a PRISMA flow diagram, which will detail each stage of the screening process, including the number of studies included and excluded, as well as the primary reasons for exclusion [52].

**2.4 Data extraction.**   Qualitative reviews focus on ensuring transparent and reliable data extraction through consensus and quality control rather than assessing the level of agreement between different reviewers [54]. Therefore, data extraction will be performed independently by one reviewer. The second reviewer will check the completeness and accuracy of the data extracted by the first reviewer, aiming to control and monitor the quality of the extraction and ensure that no important data is missed. To ensure systematic data extraction and management, a data extraction form will be designed to extract the basic characteristics and main findings of each study. Specifically, the following information will be extracted: authors, date of publication and country, data collection method (e.g., focus groups and interviews), number and type of participants recruited (including people living with dementia, family caregivers, and healthcare providers), data analysis method, and main findings. If necessary, the corresponding authors will be contacted to obtain any missing data. The extracted data will provide a structured foundation for the interpretation and synthesis of each study, facilitating a more coherent and meaningful integration of findings within the broader context of the research [55]. An additional column will be included in the data extraction form to record the name or identifier of the data extractor and the date of data extraction for each included study, thereby ensuring transparency and traceability in the data extraction process. A draft of the data extraction tool is provided in S3 File.

**2.5 Quality appraisal.**   The quality of the studies will be assessed using the CASP (Critical Appraisal Skills Programme) checklists [56] (Table 2) and completed independently by two reviewers. The CASP tool, which is widely recognized and recommended for qualitative research in health studies [57]. Any disagreement between reviewers on the CASP assessment will be discussed until consensus is reached. If necessary, a third reviewer will be consulted if the two reviewers cannot reach a consensus. In addition, a table will be provided to clearly

**Table 2. Summary of the CASP critical appraisal criteria [56].**

| CASP critical appraisal criteria |
| --- |
| 1. Was there a clear statement of the aims of the research? |
| 2. Is a qualitative methodology appropriate? |
| 3. Was the research design appropriate to address the aims of the research? |
| 4. Was the recruitment strategy appropriate to the aims of the research? |
| 5. Were the data collected in a way that addressed the research issue? |
| 6. Has the relationship between researcher and participants been adequately considered? |
| 7. Have ethical issues been taken into consideration? |
| 8. Was the data analysis sufficiently rigorous? |
| 9. Is there a clear statement of findings? |
| 10. Is the research valuable to clinical practice? |

present the detailed quality assessment of each included study, containing the specific assessment results of each study in each CASP assessment domain, to ensure transparency in the quality assessment process. The purpose of using the CASP appraisal process is not to exclude papers before synthesis, but rather to evaluate their contributions at a later stage, and make informed judgments about their overall impact on the research synthesis [58].

## Phase 4–7: Relationship, translation, and synthesis of studies

**2.6 Data synthesis and analysis.** Meta-ethnography will be chosen to synthesize the findings of the included studies [59]. Data synthesis and analysis will be based on Phase 4 to Phase 7 of the seven-stage meta-ethnographic synthesis proposed by Noblit and Hare (1988) [46].

*Determining how the studies are related (Phase 4).* The core of this stage is to identify the connections between different studies, especially to analyze the interrelationships between their key constructs. Meta-ethnography widely uses Schutz's (1971) concepts of first-order, second-order, and third-order constructs for analysis [60]. First-order constructs are original findings or data, i.e., participants' views or interpretations of personal experiences, usually in the form of quotes. Second-order constructs refer to the original study author's interpretation of the participants' views, the themes, conclusions, interpretations, and recommendations drawn by the original researchers will form the basis for the second-order constructs. Third-order constructs are the interpretations that the review team draws from first-order and second-order constructs, that is, a higher level of understanding generated by comparing, integrating, and reinterpreting second-order constructs from multiple studies [47]. Definitions of first-order, second-order, and third-order constructs in this review are listed in Table 3.

Meta-ethnography focuses on "second-order constructs", that is, constructing an analytical framework from the original researcher's interpretation rather than directly extracting themes from the original data [46]. Therefore, the specific operations of this stage include three core steps [61] (i) Coding: The primary reviewer will code the results section of all included studies to identify second-order constructs (the original researcher's interpretation of the participants' views), expressed in terms of themes and concepts, and extract first-order constructs (the participants' original views) to support in-depth interpretation of the second-order constructs. Two additional reviewers will review the second-order constructs of each study to reduce bias and errors in coding. (ii) Comparison of second-order constructs across studies. Following the recommendations of Noblit and Hare (1988), a theme list will be created for each study to systematically explore and compare the similarities and differences of second-order constructs between different studies [46]. (iii) Identify the relationships between second-order constructs across studies. By systematically sorting out the relationships between constructs, a clear direction will be provided for subsequent translation and synthesis.

*Translating the studies into one another (phase 5).* The process of "translation" within meta-ethnography transcends mere literal meaning; it involves a careful understanding and

**Table 3. Definitions of first-order, second-order, and third-order constructs in this review.**

| Constructs | Definitions |
| --- | --- |
| First-order constructs | Perceptions or experiences of people living with dementia, family caregivers, or healthcare providers regarding home-based end-of-life care for people living with dementia. |
| Second-order constructs | The primary authors' interpretation of participants' experiences or perceptions of home-based end-of-life care for people living with dementia (expressed in terms of themes and concepts). |
| Third-order constructs | Interpretations of first-order and second-order constructs by reviewers of this review (expressed in terms of themes and concepts). |

synthesis of findings within the different contextual frameworks found across the studies [62]. This "translation" process is key to meta-ethnography and involves comparative analysis of second-order constructs across studies to identify points of convergence and divergence between findings [46].

When different studies have similar constructs (i.e., themes), a method of reciprocal translation is employed to promote their integration. Conversely, if constructs across studies present contentious points, a process of refutational translation is initiated to clarify and synthesize these differences [63]. The core of refutational translation is to deeply examine the root causes of these disputes, exploring factors such as research designs, sample characteristics, and the ideological positions of researchers [58]. Given the nature of meta-ethnography, both reciprocal and refutational translations may be applied concurrently [58].

Specifically, comparing the second-order constructs from the first study with the constructs from the second study to identify similarities and differences, with similar constructs being merged. Then compare the merged constructs from the two studies with the second-order constructs that emerged from the third study, identifying similarities, note any differences as well as any additional insights that the third study contributes to the constructs. This process will continue in sequence until the second-order constructs of all studies have been translated into each other [64, 65]. It is essential to note that the order in which studies are compared may significantly influence the final integrated outcomes [66]. Given the ongoing development of theoretical frameworks, policies, and care practices for home-based end-of-life care for people living with dementia over time, the comparison and synthesis of studies will be in order of publication time, starting with the earliest studies [67]. This will allow for a better understanding of the evolution of challenges and facilitators of home-based end-of-life care and how these factors have changed or persisted over time.

In addition, this review will examine the perspectives and experiences of different stakeholders on home-based end-of-life care for people living with dementia, so in order to avoid misunderstandings caused by mixing perspectives from different populations, studies from different populations will be translated and analyzed separately. Consequently, the translation process will include three stages: (i) reciprocal or refutational translation of studies with "people living with dementia" as participants to understand their views on home-based end-of-life care; (ii) reciprocal or refutational translation of studies with "family caregivers" as participants to understand their views and experiences related to the home-based end-of-life care for people living with dementia; (iii) reciprocal or refutational translation of studies with "healthcare providers" as participants to identify the challenges and opportunities they encounter while delivering home-based end-of-life care for people living with dementia. Translating and synthesizing studies by population will contribute to the development of a more nuanced interpretive framework, highlighting points of convergence and divergence among different participants' perspectives on the barriers and facilitators of home-based end-of-life care for people living with dementia.

It is important to note that this review did not consider group translation synthesis based on geographical or cultural backgrounds. While geographical and cultural contexts significantly influence participants' perspectives and experiences, the relatively small scale of current research on home-based end-of-life care for people living with dementia means that further subdivision by these factors could result in some subgroups having insufficient studies to synthesis and draw meaningful conclusions. Nevertheless, the impact of contextual factors on the findings will be considered throughout the review, especially geographical and cultural contexts, to clarify their role in shaping participants' understanding and experiences of home-based end-of-life care services.

*Synthesizing translations (Phase 6).* Synthesizing translations is described as transforming the whole into something that possesses greater meaning than the individual parts [68]. This phase will be divided into the following core steps: Develop third-order constructs, argumentative synthesis, and construct interpretive frameworks.

Third-order constructs (reviewers' further interpretations of second-order constructs) [47] will first be created by synthesizing and interpreting second-order constructs identified in studies related to people living with dementia, family caregivers, and healthcare providers respectively. To ensure nuanced and rich interpretations while minimizing bias and error during data analysis and synthesis, third-order constructs will be developed by one reviewer and then discussed in depth with the second and third reviewers until consensus on the third-order constructs is reached.

The third-order constructs for each participant group will be linked into a cohesive "thread of argument". Through this argumentative thread, the first synthesis is implemented, that is, to provide an independent interpretive structure for each participant group. Subsequently, the three independent interpretive structures will be merged into a final argumentative synthesis. Argumentative synthesis is a systematic method for comprehensive analysis of qualitative research, which can collect and interpret the views of two or more different groups and explain the relationship between themes to form a complete and consistent interpretive framework [69]. The studies of each participant group in this review focused on the facilitators and barriers of home-based end-of-life care for people living with dementia, which can achieve effective integration and construct a comprehensive argument about this phenomenon.

At the end of this phase, a multidimensional framework will be developed to explain the facilitators and barriers to home-based end-of-life care for people living with dementia. This framework will consider factors at various participant levels and emphasize the interconnections among these elements, deepening the understanding of the key driving conditions and barriers involved in the home-based end-of-life care process for people living with dementia.

*Expressing the synthesis (Phase 7).* The final phase of this review includes (i) summarizing the main findings of the review; (ii) reflecting on and describing the strengths and limitations of this review and the impact of these on the synthesis process and results; and (iii) based on the review results, proposing feasible follow-up research designs or practical work suggestions for unresolved issues in the field of home-based end-of-life care for people living with dementia [51].

## 3. Discussion

People living with dementia want to spend their final days at home. However, the gap between this ideal and the actual place where they ultimately die (hospital or care facility) is worth considering. Because the fulfillment of patients' end-of-life wishes is an important indicator of quality of death [70]. While some qualitative studies have explored factors that facilitate and challenge of home- based end-of-life care for people living with dementia, these findings are scattered and lack a systematic synthesis. The fragmentation of evidence hinders a comprehensive understanding of how to optimize end-of-life care at home for people living with dementia. To address this issue, this review will systematically synthesize the existing qualitative research evidence using a meta-ethnography approach. Meta-ethnography, as a specific qualitative synthesis method, reinterprets and translates existing research results, generate new interpretations and insights beyond a single study [48]. To our knowledge, this review will be the first meta-ethnography to comprehensively explore the facilitators and barriers to home-based end-of-life care for people living with dementia, and it also marks the first attempt to construct an interpretive framework in this context. By focusing on the perspectives of a range

of key stakeholders closely connected with this phenomenon (people living with dementia, family caregivers, and healthcare providers), it will produce evidence that will contribute to a comprehensive understanding of the barriers and facilitators of end-of-life care for people living with dementia in the home setting. This evidence may help to narrow the long-standing evidence-practice gap in the field of home-based end-of-life care for people living with dementia.

Importantly, this review aims not only to deepen theoretical understanding in the field but also to provide significant practical implications for policy and practice. A comprehensive analysis of facilitating factors will provide valuable insights for optimizing and innovating current models of home-based end-of-life care for people living with dementia, ensuring that services are acceptable to people living with dementia and their family caregivers, as well as feasible for healthcare providers. Identifying barriers to home-based end-of-life care for people living with dementia will provide a critical reference for developing precise and effective intervention strategies to address practical difficulties. Consequently, the findings of this review will help provide an evidence base to guide policymakers and practitioners to optimize relevant policies and services, and can take into account the needs and challenges of different stakeholders when formulating and implementing home-based end-of-life care strategies for people living with dementia, so as to better support people living with dementia to achieve their wishes of dying at home in policies and services.

Despite the importance and impact of this review, potential limitations of this review must be acknowledged. The meta-ethnography approach, while offering in-depth reinterpretations, may be influenced by the subjective interpretation of data. Additionally, the use of a single reviewer to code the second-order constructs and develop the third-order constructs represents a limitation of this protocol. Ideally, team members should independently code second-order constructs and develop the third-order constructs before discussing and these constructs to generate an "argumentative" synthesis [47]. However, in practice, this approach may present challenges. Given that the process of synthesis and translation is inherently interpretive, different researchers may produce different interpretations, and differences in interpretation between team members may make the integration of results complicated and difficult. Therefore, this review decided to use a single reviewer to code the second-order constructs and develop the third-order constructs to ensure uniformity and consistency in the data analysis process. However, to maintain the rigor of the findings, the second and third reviewers will be invited to review the second-order constructs and third-order constructs developed. This auditing phase not only provides additional perspectives but also effectively mitigates potential biases introduced by a single coder, thereby enhancing the reliability of the research results. In addition, to reduce the risks associated with subjective interpretation, a transparent analytical process will be employed each step of data analysis and theme extraction will be reported in detail, along with sufficient information in the report to facilitate understanding and evaluation of the data interpretation process.

Additionally, it should be recognized that the value of home-based end-of-life care is deeply rooted in specific cultural contexts and is influenced by various factors, including social and healthcare structural differences [71, 72]. This review will draw meaningful conclusions from studies conducted in countries with different cultural backgrounds, health systems and settings [73]. Therefore, when comparing research findings, particular attention will be paid to these contextual factors in order to identify and analyze their potential impact on the results. Furthermore, any discrepancies between research outcomes and conclusions that arise due to cultural differences will be carefully examined to avoid drawing misleading conclusions. This will ensure the validity and applicability of the insights drawn from the review across different cultures and settings.

It should also be acknowledged that this review only includes studies published in English due to a lack of available translation resources. In addition, the grey literature lacks standardization, and its research quality and reliability cannot be guaranteed, which will also be excluded. Although the exclusion of grey literature and non-English studies can improve the quality and efficiency of the review, it may inadvertently limit the comprehensiveness and cultural diversity of the review results. Because focusing only on English literature may introduce knowledge bias and ignore important findings published in other languages, and limit the understanding of the diversity and complexity of home-based end-of-life care practices for people living with dementia around the world. In particular, there may be significant differences in the concept and practice of home-based end-of-life care in different cultural contexts. Grey literature may contain rich practical experience and innovative cases, which are essential for understanding the specific needs and barriers of people living with dementia in home-based end-of-life care. Therefore, excluding non-English studies and grey literature may limit the breadth of the review and hinder a comprehensive understanding of the facilitators and barriers to home-based end-of-life care for people living with.

However, it is worthy of recognition that although this review has certain limitations, several measures have been adopted to ensure methodological rigor, scientific merit and credibility of the results. To ensure the credibility of the results of this review, this review will adopt a rigorous review method, including clear search procedures, screening criteria and data extraction to maximize review validity and transparency. This review report will also be structured using the meta-ethnography reporting guidelines (eMERGe) and PRISMA statement guidelines to ensure the clarity and completeness of this review report. A rigorous and transparent review process and methodology will enhance the credibility of this review's results and contribute to its potential influence in promoting policy and improving practice.

Ethical considerations will also be carefully considered in this review, based on the recognition that sensitive data may be involved. Although this review will be based on existing literature and the data are publicly available, the participants are often vulnerable people living with dementia and their family caregivers; the data may also contain sensitive information such as personal experiences and mental health conditions. In addition, the experiences of people living with dementia and their family caregivers in the end-of-life care process often involve complex ethical and cultural factors, such as end-of-life decision-making. Therefore, relevant data will be cited and integrated with care and sensitivity to avoid any threat or misunderstanding to the privacy and dignity of participants and to ensure that no identifiable individual information is exposed. This prudence is not only a respect for the privacy of the participants, but also a reflection of ensuring the ethical responsibility of the review.

In summary, this review is significant as the first comprehensive meta-ethnography to explore the facilitators and barriers of home-based end-of-life care for people living with dementia, thereby constructing an interpretive framework that encompasses diverse perspectives from people living with dementia, family caregivers, and healthcare providers. The review findings will help bridge the significant gap between the expressed desire of people living with dementia to die at home and the reality of where they ultimately pass away, typically in hospitals or care facilities. The findings will also serve as an evidence base for guiding policymakers and practitioners in developing home-based end-of-life care services and policies, ultimately enhancing support for people living with dementia in achieving their wish to die at home.

## Supporting information

**S1 File. MEDILINE search strategy.**
(DOCX)

**S2 File. PRISMA-P checklist.**
(DOCX)

**S3 File. Data extraction tools.**
(DOCX)

## Author Contributions

**Conceptualization:** Guo Yin.

**Formal analysis:** Guo Yin, Divya Sivaramakrishnan, Leah Macaden.

**Resources:** Leah Macaden.

**Supervision:** Divya Sivaramakrishnan, Leah Macaden.

**Visualization:** Leah Macaden.

**Writing – original draft:** Guo Yin.

**Writing – review & editing:** Divya Sivaramakrishnan, Leah Macaden.

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
