## [Decision Letter · Decision Letter 0]

7 Oct 2024

PONE-D-24-40452Bring dying at home: What facilitates and hinders home-based end-of-life care for people living with dementia? - A systematic review and meta-ethnography protocolPLOS ONE

Dear Dr. Yin,

Thank you for submitting your manuscript to PLOS ONE. After careful consideration, we feel that it has merit but does not fully meet PLOS ONE’s publication criteria as it currently stands. Therefore, we invite you to submit a revised version of the manuscript that addresses the points raised during the review process.

We look forward to receiving your revised manuscript.

Kind regards,

Mostafa Shaban

Academic Editor

PLOS ONE

Journal Requirements:

2. As required by our policy on Data Availability, please ensure your manuscript or supplementary information includes the following: A numbered table of all studies identified in the literature search, including those that were excluded from the analyses. For every excluded study, the table should list the reason(s) for exclusion. If any of the included studies are unpublished, include a link (URL) to the primary source or detailed information about how the content can be accessed. A table of all data extracted from the primary research sources for the systematic review and/or meta-analysis. The table must include the following information for each study: Name of data extractors and date of data extraction Confirmation that the study was eligible to be included in the review. All data extracted from each study for the reported systematic review and/or meta-analysis that would be needed to replicate your analyses. If data or supporting information were obtained from another source (e.g. correspondence with the author of the original research article), please provide the source of data and dates on which the data/information were obtained by your research group. If applicable for your analysis, a table showing the completed risk of bias and quality/certainty assessments for each study or outcome. Please ensure this is provided for each domain or parameter assessed. For example, if you used the Cochrane risk-of-bias tool for randomized trials, provide answers to each of the signalling questions for each study. If you used GRADE to assess certainty of evidence, provide judgements about each of the quality of evidence factor. This should be provided for each outcome. An explanation of how missing data were handled. This information can be included in the main text, supplementary information, or relevant data repository. Please note that providing these underlying data is a requirement for publication in this journal, and if these data are not provided your manuscript might be rejected.

Reviewers' comments:

Reviewer's Responses to Questions

**Comments to the Author**

1. Does the manuscript provide a valid rationale for the proposed study, with clearly identified and justified research questions?

Reviewer #1: Yes

Reviewer #2: Yes

2. Is the protocol technically sound and planned in a manner that will lead to a meaningful outcome and allow testing the stated hypotheses?

Reviewer #1: Yes

Reviewer #2: Yes

3. Is the methodology feasible and described in sufficient detail to allow the work to be replicable?

Reviewer #1: Yes

Reviewer #2: Yes

4. Have the authors described where all data underlying the findings will be made available when the study is complete?

Reviewer #1: No

Reviewer #2: No

5. Is the manuscript presented in an intelligible fashion and written in standard English?

Reviewer #1: Yes

Reviewer #2: Yes

6. Review Comments to the Author

You may also provide optional suggestions and comments to authors that they might find helpful in planning their study.

Reviewer #1: Strengths

Relevant Topic: The article addresses a critical issue in healthcare—home-based end-of-life care for people living with dementia, which is timely and significant.

Methodological Rigor: The use of meta-ethnography to synthesize qualitative studies is appropriate for the research question and allows for a comprehensive understanding of the facilitators and barriers involved.

Areas for Improvement

Clarification of Methodology: Request clearer descriptions of the meta-ethnography process, including how studies will be selected and analyzed.

Addressing Limitations: Suggest the authors explicitly discuss potential limitations of their review, such as the exclusion of non-English studies or the impact of cultural factors on the findings.

Expected Outcomes: Encourage the authors to specify the expected contributions of their review to both practice and policy, enhancing the practical implications of their research.

Editorial Suggestions

Improve clarity and coherence in certain sections.

Ensure all terms and concepts are well-defined for broader accessibility.

Reviewer #2: 1. Revise the introduction to clearly justify why this review is necessary.

2. Identify gaps in the literature and articulate how this study provides new insights.

3. Provide detailed steps on how each of Noblit and Hare’s phases will be implemented.

4. Clarify how "reciprocal" and "refutational" translations will be determined.

5. Include a detailed search strategy with specific terms, and justify the exclusion of non-English studies.

6. Clearly explain how the eMERGe and PRISMA guidelines will be applied at each stage of the review.

7. Specify their role in structuring the analysis and reporting.

8. Expand the discussion on bias, particularly how the subjective nature of translation and synthesis will be mitigated.

9. Propose steps such as independent coding or triangulation to minimize researcher influence.

10. Include a section explaining how cultural differences will be accounted for in the synthesis.

11. Consider conducting subgroup analyses based on geographical or cultural context.

12. Add a section discussing ethical considerations, especially how you will handle sensitive data from the reviewed studies.

13. Provide a strong justification for excluding grey literature and non-English studies.

14. Discuss how these exclusions may limit the comprehensiveness and cultural diversity of the findings

7. PLOS authors have the option to publish the peer review history of their article (what does this mean?). If published, this will include your full peer review and any attached files.

Reviewer #1: No

Reviewer #2: No

---

## [Author Response · Author response to Decision Letter 0]

29 Oct 2024

Dear Editors and Reviewers: 

We are very grateful to you for giving us an opportunity to revise our manuscript. We also deeply appreciate your positive and constructive comments and suggestions on our manuscript entitled: “Bring dying at home: What facilitates and hinders home-based end-of-life care for people living with dementia? - A systematic review and meta-ethnography protocol”. These comments are all valuable and helpful for improving the quality of our manuscript. 

We have considered the reviewers’ comments carefully and tried our best to revise the manuscript. These changes will not influence the content and framework of this manuscript. The relevant changes in the revised manuscript are marked in red. 

We sincerely appreciate the warm effort of the editors and reviewers, and hope that the correction will meet with approval. Thank you again for your comments and suggestions. 

Reviewer #1: 

Comments 1:

Strengths

Relevant Topic: The article addresses a critical issue in healthcare—home-based end-of-life care for people living with dementia, which is timely and significant.

Methodological Rigor: The use of meta-ethnography to synthesize qualitative studies is appropriate for the research question and allows for a comprehensive understanding of the facilitators and barriers involved.

Response: Thank you so much for your recognition of this review protocol.

Comments 2 : Clarification of Methodology: Request clearer descriptions of the meta-ethnography process, including how studies will be selected and analysed.

Response: Thank you very much for your suggestions. Your feedback has greatly improved the clarity of our work. The main revisions are in Sections 2.3 & 2.6 of the manuscript. We have also added a figure (Figure 1, Page 12) illustrating the seven stages of meta-ethnography to better demonstrate its application in this review.

Comments 3 :Addressing Limitations: Suggest the authors explicitly discuss potential limitations of their review, such as the exclusion of non-English studies or the impact of cultural factors on the findings.

Response: Thank you for your suggestions. Your feedback has been invaluable in strengthening this section. We have added a discussion of the potential limitations of the review in the Discussion section, including the exclusion of non-English studies and the impact of cultural factors on our findings(Line 584-614, Page31-32 ). 

“Additionally, it should be recognized that the value of home-based end-of-life care is deeply rooted in specific cultural contexts and is influenced by various factors, including social and healthcare structural differences (58, 59). This review will draw conclusions from studies conducted in countries with different cultural backgrounds, health systems and settings (60). Therefore, when comparing research findings, particular attention will be paid to these contextual factors in order to identify and analyse their potential impact on the results. Furthermore, any discrepancies between research outcomes and conclusions that arise due to cultural differences will be carefully examined to avoid drawing misleading conclusions. This will ensure the validity and applicability of the insights drawn from the review across different cultures and settings.”

“It should also be acknowledged that this review only includes studies published in English due to a lack of available translation resources. In addition, the grey literature lacks standardization, and its research quality and reliability cannot be guaranteed, which will also be excluded. Although the exclusion of grey literature and non-English studies can improve the quality and efficiency of the review, it may inadvertently limit the comprehensiveness and cultural diversity of the review results. Because focusing only on English literature may introduce knowledge bias and ignore important findings published in other languages, and limit the understanding of the diversity and complexity of home-based end-of-life care practices for people living with dementia around the world. In particular, there may be significant differences in the concept and practice of home-based end-of-life care in different cultural contexts. Grey literature may contain rich practical experience and innovative cases, which are essential for understanding the specific needs and barriers of people living with dementia in home-based end-of-life care. Therefore, excluding non-English studies and grey literature may limit the breadth of the review and hinder a comprehensive understanding of the facilitators and barriers to home-based end-of-life care for people living with.”

Comments 4 :Expected Outcomes: Encourage the authors to specify the expected contributions of their review to both practice and policy, enhancing the practical implications of their research.

Response: We appreciate your suggestions. We have improved this section in the Discussion section to clarify the practical implications of this review (Line 530-557, Page 28-30).

Reviewer #2: 

Comments 1 & Comments 2. Revise the introduction to clearly justify why this review is necessary; Identify gaps in the literature and articulate how this study provides new insights.

Response: Thank you for your suggestions, which have greatly improved the clarity of the manuscript. We have revised the Introduction to clarify the necessity and importance of this review by identifying existing gaps in the literature (Line 151-174, Page 8-9).

Comments 3. Provide detailed steps on how each of Noblit and Hare’s phases will be implemented.

Response: Thank you for your suggestions. We have added detailed steps outlining how each of Noblit and Hare’s phases will be implemented in our revised manuscript. We have also added a figure illustrating the seven stages of meta-ethnography to better demonstrate its application in this review (Figure1, Page 12).

Comments 4. Clarify how “reciprocal” and “refutational” translations will be determined.

Response: Thank you for your comments. We have added clarification on how “reciprocal” and “refutational” translations will be determined in the revised manuscript (Line 413-421,Page 23).

“When different studies have the similar constructs (i.e., themes), a method of reciprocal translation is employed to promote their integration. Conversely, if constructs across studies present contentious points, a process of refutational translation is initiated to clarify and synthesize these differences(63). The core of refutational translation is to deeply examine the root causes of these disputes, exploring factors such as research designs, sample characteristics, and the ideological positions of researchers(58).”

Comments 5. Include a detailed search strategy with specific terms, and justify the exclusion of non-English studies.

Response: Thank you for your comments. We have included a detailed search strategy with specific terms in the manuscript (Table 1 presents a detailed summary of the search terms. The MEDILINE search strategy is also provided in S1 File). Additionally, we have provided a justification for the exclusion of non-English studies (Line 295-298,Page 16).

“2.2.2 Exclusion criteria : Non-English literature. The main reason for excluding non-English literature is that language barriers may lead to misunderstanding and inaccurate interpretation of research findings. Resource limitations also make it impossible to use formal translation services. ”

Comments 6&7. Clearly explain how the eMERGe and PRISMA guidelines will be applied at each stage of the review; Specify their role in structuring the analysis and reporting.

Response: Thank you for your insightful comments. We have now included a clear explanation of how the eMERGe and PRISMA guidelines will be applied at each stage of the review, specifying their roles in structuring the analysis and reporting. (Line 228-239, Page 11-12).

“This review report will also be presented in a structured manner using the Meta-Ethnography Reporting Guidelines (eMERGe) (50) and the PRISMA Statement Guidelines (51) to ensure the clarity and completeness of this review report. Specifically, the eMERGe guidelines will guide this review to provide a detailed description of the seven stages of the meta-ethnographic process, including study selection, data extraction, translation, and synthesis. Meanwhile, the PRISMA statement guidelines will be utilized to ensure that the methodological rigor, transparency, and reliability of this systematic review, including clearly defined research questions, comprehensive literature searches, selection criteria, data extraction methods, and result reporting. The integration of these two guidelines will help present the process and results of this review in a more systematic way.”

Comments 8 & 9. Expand the discussion on bias, particularly how the subjective nature of translation and synthesis will be mitigated; Propose steps such as independent coding or triangulation to minimize researcher influence.

Response: Thank you for your insightful comments. We have expanded the discussion on bias in the manuscript, specifically addressing the subjective nature of translation and synthesis (Line 560-582, Page 30-31).

Comments 10. Include a section explaining how cultural differences will be accounted for in the synthesis.

Response: Thank you for your suggestions. We have added a section to explain how we account for cultural differences in the synthesis(Line 584-595, Page 31).

“Additionally, it should be recognized that the value of home-based end-of-life care is deeply rooted in specific cultural contexts and is influenced by various factors, including social and healthcare structural differences (58, 59). This review will draw meaningful conclusions from studies conducted in countries with different cultural backgrounds, health systems and settings (60). Therefore, when comparing research findings, particular attention will be paid to these contextual factors in order to identify and analyse their potential impact on the results. Furthermore, any discrepancies between research outcomes and conclusions that arise due to cultural differences will be carefully examined to avoid drawing misleading conclusions. This will ensure the validity and applicability of the insights drawn from the review across different cultures and settings.”

Comments 11. Consider conducting subgroup analyses based on geographical or cultural context. 

Response: Thank you very much for your valuable suggestions. We have carefully considered your suggestions and ultimately decided not to conduct subgroup analyses based on geographic or cultural backgrounds. The reasons are also included in the manuscript (Line 459-469, Page 25).

“It is important to note that this review did not consider group translation synthesis based on geographical or cultural backgrounds. While geographical and cultural contexts significantly influence participants’ perspectives and experiences, the relatively small scale of current research on home-based end-of-life care for people living with dementia means that further subdivision by these factors could result in some subgroups having insufficient studies to synthesis and draw meaningful conclusions. Nevertheless, the impact of contextual factors on the findings will be considered throughout the review, especially geographical and cultural contexts, to clarify their role in shaping participants’ understanding and experiences of home-based end-of-life care services.”

Comments 12. Add a section discussing ethical considerations, especially how you will handle sensitive data from the reviewed studies.

Response: Thank you for your suggestions. We have added a section discussing ethical considerations in the Discussion section, specifically addressing how we will manage sensitive data from the reviewed studies (Line 628-641, Page 33-34).

“Ethical considerations will also be carefully considered in this review, based on the recognition that sensitive data may be involved. Although this review will be based on existing literature and the data are publicly available, the participants are often vulnerable people living with dementia and their family caregivers, and the data may also contain sensitive information such as personal experiences and mental health conditions. In addition, the experiences of people living with dementia and their family caregivers in the end-of-life care process often involve complex ethical and cultural factors, such as end-of-life decision-making. Therefore, relevant data will be cited and integrated with care and sensitivity to avoid any threat or misunderstanding to the privacy and dignity of participants and to ensure that no identifiable individual information is exposed. This prudence is not only a respect for the privacy of the participants, but also a reflection of ensuring the ethical responsibility of the review.”

Comments 13 &14. Provide a strong justification for excluding grey literature and non-English studies; Discuss how these exclusions may limit the comprehensiveness and cultural diversity of the findings.

Response: Thank you for your insightful comments. We have added a strong justification for excluding grey literature and non-English studies in the revised manuscript. Additionally, we have discussed the potential impact of these exclusions on findings(Line 597-614, Page 32).

“It should also be acknowledged that this review only includes studies published in English due to a lack of available translation resources. In addition, the grey literature lacks standardization, and its research quality and reliability cannot be guaranteed, which will also be excluded. Although the exclusion of grey literature and non-English studies can improve the quality and efficiency of the review, it may inadvertently limit the comprehensiveness and cultural diversity of the review results. Because focusing only on English literature may introduce knowledge bias and ignore important findings published in other languages, and limit the understanding of the diversity and complexity of home-based end-of-life care practices for people living with dementia around the world. In particular, there may be significant differences in the concept and practice of home-based end-of-life care in different cultural contexts. Grey literature may contains rich practical experience and innovative cases, which are essential for understanding the specific needs and barriers of people living with dementia in home-based end-of-life care. Therefore, excluding non-English studies and grey literature may limit the breadth of the review and hinder a comprehensive understanding of the facilitators and barriers to home-based end-of-life care for people living with.”

Editorial Suggestions:

1. Improve clarity and coherence in certain sections.

Response: Thank you so much for your suggestions. We reviewed the manuscript again and revised some sections to improve clarity and coherence. Your suggestions are very valuable in improving the manuscript.

2.Ensure all terms and concepts are well-defined for broader accessibility.

Response: Thank you so much for your feedback. We have made sure to define all key terms and concepts throughout the manuscript to enhance broader accessibility.

---

## [Decision Letter · Decision Letter 1]

12 Dec 2024

Bring dying at home: What facilitates and hinders home-based end-of-life care for people living with dementia? - A systematic review and meta-ethnography protocol

PONE-D-24-40452R1

Dear Dr. Yin,

We’re pleased to inform you that your manuscript has been judged scientifically suitable for publication and will be formally accepted for publication once it meets all outstanding technical requirements.

Kind regards,

Mostafa Shaban

Academic Editor

PLOS ONE

Additional Editor Comments (optional):

Reviewers' comments:

Reviewer's Responses to Questions

**Comments to the Author**

1. Does the manuscript provide a valid rationale for the proposed study, with clearly identified and justified research questions?

Reviewer #3: Yes

Reviewer #4: Yes

2. Is the protocol technically sound and planned in a manner that will lead to a meaningful outcome and allow testing the stated hypotheses?

Reviewer #3: Yes

Reviewer #4: Yes

3. Is the methodology feasible and described in sufficient detail to allow the work to be replicable?

Reviewer #3: Yes

Reviewer #4: Yes

4. Have the authors described where all data underlying the findings will be made available when the study is complete?

Reviewer #3: Yes

Reviewer #4: Yes

5. Is the manuscript presented in an intelligible fashion and written in standard English?

Reviewer #3: Yes

Reviewer #4: Yes

6. Review Comments to the Author

You may also provide optional suggestions and comments to authors that they might find helpful in planning their study.

Reviewer #3: Dear Authors,

I would like to commend you on the thorough revisions you made to the manuscript. You have addressed all of the concerns and suggestions from the initial review in a comprehensive manner, and the changes have greatly enhanced the overall quality of the work. The manuscript is now much clearer, and the improvements in the methodology and discussion sections, in particular, have strengthened the scientific rigor of the study. I believe the revised version is significantly better, and I appreciate the effort you have put into improving the manuscript.

Reviewer #4: Dear authors,

Many thanks for the prompt answer and response to the suggestions. I hope success for you.

Regards

7. PLOS authors have the option to publish the peer review history of their article (what does this mean?). If published, this will include your full peer review and any attached files.

Reviewer #3: **Yes: **Enas Abdelaziz

Reviewer #4: No

---

## [Editor Report · Acceptance letter]

16 Dec 2024

PONE-D-24-40452R1 

PLOS ONE

Dear Dr. Yin, 

I'm pleased to inform you that your manuscript has been deemed suitable for publication in PLOS ONE. Congratulations! Your manuscript is now being handed over to our production team.

Kind regards, 

on behalf of

Dr. Mostafa Shaban 

Academic Editor

PLOS ONE